# Metabolic Features of Brain Function with Relevance to Clinical Features of Alzheimer and Parkinson Diseases

**DOI:** 10.3390/molecules27030951

**Published:** 2022-01-30

**Authors:** David Allan Butterfield, Maria Favia, Iolanda Spera, Annalisa Campanella, Martina Lanza, Alessandra Castegna

**Affiliations:** 1Department of Chemistry, University of Kentucky, Lexington, KY 40506, USA; 2Sanders-Brown Center on Aging, University of Kentucky, Lexington, KY 40536, USA; 3Department of Biosciences, Biotechnologies and Biopharmaceutics, University of Bari, 70125 Bari, Italy; maria.favia@uniba.it (M.F.); iolanda.spera@uniba.it (I.S.); annalisa.campanella@uniba.it (A.C.); martina.lanza@uniba.it (M.L.)

**Keywords:** Alzheimer’s disease, Parkinson’s disease, AD, PD brain metabolism, glucose, metabolic reprogramming, neurodegeneration, oxidative stress

## Abstract

Brain metabolism is comprised in Alzheimer’s disease (AD) and Parkinson’s disease (PD). Since the brain primarily relies on metabolism of glucose, ketone bodies, and amino acids, aspects of these metabolic processes in these disorders—and particularly how these altered metabolic processes are related to oxidative and/or nitrosative stress and the resulting damaged targets—are reviewed in this paper. Greater understanding of the decreased functions in brain metabolism in AD and PD is posited to lead to potentially important therapeutic strategies to address both of these disorders, which cause relatively long-lasting decreased quality of life in patients.

## 1. Introduction

Recent studies highlight the importance of metabolism in the regulation of brain function, with the discovery of metabolism-linked genes and functional states. This may provide clues to the understanding of how metabolism influences the onset and progression of neurodegeneration. To this end, it is essential to comprehend the metabolic specificities of brain function, and to develop tools to dissect the metabolic pathways potentially involved in loss of brain function.

Brain metabolism represents 20% of the body’s total oxygen consumption; it is highly dynamic, as brain metabolism responds to dynamic energy consumption typical of the central nervous system (CNS). Energy depletion triggers compensatory mechanisms to enhance both metabolism and oxygen availability, and this occurs in a region-specific fashion and within the same region, via specific neuronal structures (i.e., synapses). Neurons are the main utilizers of the energy produced [1], which is channeled at the synaptic level to restore membrane potential after depolarization [2].

Other energy-consuming functions include the high metabolic rates typical of neurons, axonal transport, and neurotransmitter synthesis [3,4]. It follows that energy consumption varies locally depending on neuronal function, and this variation requires a high degree of plasticity in modulating oxygen supply, mitochondrial function, and metabolism. Dysregulation of these events is consistently associated with neurodegenerative disorders [5,6,7,8,9]. In the effort to understand the metabolic mechanism(s) underlying brain function, we will start by describing the metabolic features of the brain in physiological conditions, and then discuss how metabolic changes are involved in Alzheimer’s disease and Parkinson’s disease.

Table 1 summarizes the metabolic alterations associated with the main clinical features of AD and PD discussed in this review.

## 2. Brain Metabolism in Physiological Conditions

### 2.1. Glucose Metabolism

Glucose is the main energy substrate in the brain, as it is the principal source of ATP (see Figure 1A). In the brain, glucose is taken up from the bloodstream by specific glucose transporters to be metabolized through glycolysis. Pyruvate is then transported into the mitochondria to be channeled into the TCA cycle. Oxidation of the TCA substrate produces CO_2_ and reduces NAD^+^ and FAD^+^, which are the electron donors for the oxidative phosphorylation (OXPHOS)—an electron transfer chain driven by substrate oxidation that is coupled with the synthesis of ATP through an electrochemical transmembrane gradient. The CO_2_ produced is then removed via blood circulation and eliminated through the respiratory system.

Glucose metabolism is efficiently regulated in the brain; the first step of this regulatory mechanism entails its uptake. Among the different glucose transporters, GLUT1 is mainly expressed in astrocytes and endothelial cells, whereas GLUT3 is mainly expressed in neurons. The fact that these transporters are insulin-independent does not mean that glucose uptake is not regulated by insulin. Indeed, the insulin-dependent GLUT4 [10] is also present in several brain regions, including the hippocampus and cerebellum [11]. Insulin crosses the blood–brain barrier (BBB) through a saturable transport system and, as such, brain hormone levels only partially reflect those found in the blood [12]. Insulin’s action is dependent on the expression of the insulin receptor (IR), which is abundant in neurons [13]. Insulin signaling includes the Ras/mitogen-activated protein kinase (MAPK) and PI3K/Akt pathways.

The IR and the insulin-like growth factor 1 receptor (IGF1R) convey the signal to the insulin receptor substrate-1 (IRS-1) which, in turn, intracellularly activates PI3K—a kinase involved in several intracellular signaling transduction processes [14]. Insulin binding to the IR recruits the intracellular IRS proteins through specific phosphorylation on tyrosine residues. Tyrosine-phosphorylated IRS activates PI3Ks, involving Akt phosphorylation. PI3K/Akt modulates downstream factors, such as glycogen synthase kinase 3 (GSK3), mTOR, and forkhead box (FOX) transcription factors, regulating the brain’s cellular functions [15].

mTOR is a serine/threonine kinase. In the brain, the mTOR signaling cascade is activated by nutrients, neurotrophic factors, and neurotransmitters, and enhances protein synthesis and suppresses autophagy, contributing to normal neuronal growth by promoting their differentiation, axonal budding, regeneration, and myelination, along with the growth of dendritic spines [16]. Furthermore, mTOR-regulated processes in neurons and glial cells influence important superior physiological functions such as neuronal excitability and survival, synaptic plasticity, cognition, nutrition, and circadian rhythm control [17]. Therefore, disruption of mTOR signaling may cause neurodegeneration and abnormal neural development [16]. mTOR includes two distinct complexes called mTORC1 and -2, with several interacting proteins. Different nutritional and environmental signals activate AKT which, in turn, activates mTORC1. In addition, cellular energy status, oxygen/hypoxia, and stressors regulate mTORC1 activity. In the brain, neurotransmitters, neuromodulators, and hormones are reported to activate mTORC1 [16]. Activated mTORC1 promotes cell growth by phosphorylating substrates by enhancing anabolic processes such as mRNA translation and lipid synthesis, or by limiting catabolic processes such as autophagy, while mTORC2 promotes cell survival by activating AKT [17].

IRS-1 phosphorylation at specific serine residues disrupts PI3K/Akt coupling to IGF-1 and IR, leading to IRS-1 inactivation and degradation, which is a feature of brain insulin resistance (IRes) [18].

The MAPK branch of insulin signaling is triggered by Shc phosphorylation, leading to gene expression and cell growth [15]. IRS can be serine phosphorylated by MAPK, and this reduces its signaling [19]. Moreover, the function of IRS-1 is modulated by biliverdin reductase-A (BVR-A) [20].

Glucose is metabolized through glycolysis, which occurs in all brain cell types to different extents. Indeed the activity of phosphofructokinase (PFK), which catalyzes the conversion of fructose 6-phosphate (F6P) into fructose 1,6-bisphosphate (F1,6BP), is higher in astrocytes compared to neurons [21], although studies in rat brains suggest that the glycolytic flux in both cell types seems to be interconnected [22]. Microglia also exploit glycolysis to maintain the immune functional response [23]. In particular, enhancing glycolytic flux in the microglia promotes inflammasome activation [24]. Glucose 6-phosphate (G6P) can be channeled into the pentose phosphate pathway (PPP) [25], which promotes the two-step oxidative decarboxylation of G6P catalyzed by glucose-6-phosphate dehydrogenase (G6PD) and 6-phosphogluconate dehydrogenase (6PGD), yielding ribulose-5-phosphate (R5P). Both enzymes are NADP^+^-dependent; therefore, this process leads to the production of NADPH. Hence, glucose metabolism can be diverted from glycolysis into the PPP in order to meet the NADPH demand typical of oxygen-consuming cells. NADPH contributes to the maintenance of GSH in its reduced form [26], since the exposure of GSH to ROS leads to oxidized glutathione (GSSG), which can be replenished by the activity of the NADPH-dependent GSH reductase. In the next non-oxidative steps of PPP, Ru5P is isomerized into ribose-5-phosphate (R5P), which can enter the nucleotide biosynthetic pathway or the subsequent PPP branch, leading to R5P epimerization and synthesis of different phosphorylated sugars, including glyceraldehyde-3-phosphate (G3P) and F6P [25]. These sugars can enter glycolysis, leading to pyruvate synthesis, so the F6P and G3P pools are shared by glycolysis and the PPP. In neurons, recycling of G6P can occur from F6P, due to the high glucose phosphate isomerase (GPI-1) activity [27,28]. G6P can re-enter the PPP, leading to extra NADPH production, which is strongly demanded in the brain in order to reduce oxidized glutathione and thereby protect the brain from oxidative damage.

### 2.2. TCA Cycle

The tricarboxylic acid cycle, also known as the citric acid cycle or the Krebs cycle, is a cyclic pathway that represents a major metabolic hub for cell function [29] (see Figure 2A). Acetyl-CoA, derived from pyruvate, amino acids, or fatty acid oxidation, is channeled into a cycle of reactions that sustain (1) energy production, (2) anabolic and catabolic processes, and (3) redox balance, as follows: (1) The TCA cycle oxidizes acetyl-CoA to two molecules of CO_2_, leading to the production of ATP and the reduction of NAD^+^ and FAD^+^ to NADH and FADH_2_, which enter the electron transport chain (ETC) complex I (NADH dehydrogenase) and complex II (succinate dehydrogenase, SDH), respectively. The electron flux from complexes I and II through the ETC leads to the production of ATP by means of OXPHOS, which is coupled with the TCA cycle as it reoxidizes the coenzymes necessary for TCA function. (2) Intermediates of the TCA cycle are also sources of macromolecule synthesis and, as such, are diverted from the mitochondria to the cytosol. An example is GABA—an important neurotransmitter for synaptic plasticity [30]. The diminution of TCA metabolites can be counteracted by the so-called anaplerotic reactions that provide TCA intermediates to keep the cycle running. Examples of anaplerosis are represented by glutaminolysis-related 2-oxoglutarate (2-OG) production and by pyruvate-to-oxaloacetate conversion via pyruvate carboxylase.

A typical metabolic feature of the brain is pyruvate recycling [31] that is, pyruvate synthesis from the TCA cycle intermediates malate and oxaloacetate via malic enzymes and phosphoenolpyruvate carboxykinase (PEPCK), respectively [32,33].

OXPHOS generates reactive oxygen species (ROS) as byproducts, which cannot be completely inactivated by the antioxidant defenses that in the brain—a high-oxygen-consuming organ—are quite low. Cu,Zn-superoxide dismutase, Mn-superoxide dismutase, peroxiredoxin, and glutathione (among other moieties) represent some of the endogenous antioxidant defenses. The TCA cycle can contribute to the redox balance as reducing equivalents are produced from OXPHOS and mitochondrial NADH can be converted into NADPH [34,35]. Conversely, the redox state can impinge on energy metabolism by regulating key enzyme and respiratory chain complex activities [36,37,38,39].

### 2.3. Ketone Bodies

While the brain utilizes glucose almost exclusively as its main energy source, other substrates, such as ketone bodies and lactate, contribute to metabolism in certain circumstances, especially when glucose supply is restricted or insufficient—for example, during fasting or in low-carbohydrate diets [40,41]. Uptake of the ketone bodies occurs through the monocarboxylate carriers (MCTs), which are highly expressed within the brain [42]—mainly in neurons (MCT2, [43,44]) and astrocytes (MCT1 and 4, [42,45]). Transport of ketone bodies is strongly dependent on their circulating levels [46]. Studies on murine models indicate that MCT upregulation occurs after fasting [47] and in a ketogenic diet [48]. Both β-hydroxybutyrate and acetoacetate are reduced to acetyl-CoA in the mitochondria by an NAD^+^-dependent process that does not require ATP [49]. Sources of ketone bodies can also be endogenous, as astrocytes can degrade free fatty acids (FFAs) that cross the BBB, producing ketone bodies via a mechanism mediated by adenosine-monophosphate-activated protein kinase (AMPK) activity [50], which is triggered by low glucose levels [50] and hypoxia [51]. Astrocytes can also provide lactate of glycolytic origin, which are supplied to neurons for energy purposes [52].

### 2.4. Amino Acids

Glutamate and branched-chain amino acids (BCAAs) are the main subject of this section (see Figure 3A). BCAAs (i.e., valine, leucine, and isoleucine) are essential amino acids. The catabolism of BCAAs begins with a reversible reaction of transamination catalyzed by branched-chain aminotransferase (BCAT), of which there are two isoforms: one mitochondrial and one cytosolic. These enzymes transfer the α-amino group from BCAAs to 2-OG, using vitamin B6 as a cofactor, producing branched-chain 2-oxoacids and glutamate (Figure 3A). A mitochondrial multienzyme complex of branched-chain 2-oxoacid dehydrogenase catalyzes a series of irreversible reactions, leading to acetyl-CoA, propionyl-CoA, and succinyl-CoA, which are involved in various biochemical processes.

Early studies had shown that BCAAs readily cross the BBB in rats [53]. Absorption of BCAAs at the level of the BBB exceeds that of all other amino acids [54]. Indeed, the brain’s ability to oxidize BCAAs is approximately four times higher than that of muscles [55]. Consequently, the mammalian brain constitutes an important organ of utilization for these amino acids [56]. The continuous passage of BCAAs across the BBB is mediated by specific transport systems that control the levels of metabolites and substrate/product spatial distribution in different brain areas. These transporters exhibit substrate-specificity or preference for some amino acids. The predominant transporter that has been shown to deliver BCAAs to the brain is the sodium-independent facilitated transporter LAT1 [57,58], which allows the entry of BCAAs in exchange for intracellular glutamine [59].

In the mammalian brain, in addition to their involvement in protein synthesis and energy production, BCAAs are engaged in the metabolism of neurotransmitters [56]. Glutamate is the main excitatory neurotransmitter in the mammalian brain [60]; for optimal brain function, its concentration should be relatively constant [61]. However, as glutamate is not able to cross the BBB in considerable quantities [62], it needs to be newly synthesized from constantly available precursors such as BCAAs—efficient donors of amino groups that can be quickly transported to the brain and easily transaminated [63] at the expense of 2-OG (Figure 3A). The transamination reaction occurs in astrocytes in the vicinity of the capillaries through which the BCAAs are carried by the blood. Astrocytes release branched-chain ketoacids into the extracellular fluid, from which they enter the neurons to be converted back into BCAAs, which are released into the extracellular fluid as well conveyed to astrocytes, completing the BCAA–glutamate cycle (Figure 3A).

A role of BCAA in mediating ammonia transfer between astrocytes and neurons has been postulated, based on the finding that the brain’s branched-chain amino acid aminotransferase isozymes (BCATs) can be cytosolic or mitochondrial isoforms. However, astrocytes exclusively display the mitochondrial form, whereas neurons present the cytosolic form [64] (Figure 3A). This selective localization plays an important functional role in the shuttling of ammonium nitrogen between astrocytes and neurons [64,65].

Glutamate metabolism is crucial in the brain. Generally, glutamate links amino acids to glucose metabolism through the TCA cycle, as aminotransferases use glutamate as an ammonia donor, leading to 2-OG production. Accumulating findings have contributed to the concept of “metabolic compartmentation” of glutamate—particularly in astrocytes and neurons [66]—based on the fact that glutamine synthetase (GS) and pyruvate carboxylase are exclusively present in astrocytes [67]. This represents the so-called glutamine–glutamate cycle between neurons and astrocytes, which is mediated by sodium-coupled amino acid transporters [68] (Figure 3A). This compartmentalization is conceivably related to the fact that glutamate is an excitatory neurotransmitter. Once glutamate’s signaling role is executed, it is taken up by astrocytes, in which glutamate is converted back to glutamine in a reaction catalyzed by GS. This prevents the so-called excitotoxic effect of glutamate accumulation in the synapse. Furthermore, de novo glutamate synthesis occurs exclusively in astrocytes following pyruvate-carboxylase-dependent anaplerosis [69].

Glutamate metabolism regulates ammonia levels, as it is a concomitant co-substrate of glutamate dehydrogenase (GDH), as well as alanine (ALT) and aspartate (AST) aminotransferases (Figure 3A). In the brain, due to the high NAD^+^/NADH ratio, glutamate is deaminated to 2-OG [70]; this is important in order to replenish the TCA cycle. Glutamate is also produced through transamination. ALT and AST are both present in the brain, although ALT activity is lower than that of AST [71]. Due to its presence in both astrocytes and neurons, ALT seems to be involved in ammonium nitrogen transfer between these cells [72] (Figure 3A).

Glutamate is the precursor of the inhibitory neurotransmitter γ-aminobutyric acid (GABA), via glutamate decarboxylase (GAD) [73], which in the brain consists of several isoforms exclusively present in GABAergic neurons [74]. GABA is metabolized to succinate via the enzyme succinic semialdehyde dehydrogenase (GABA-shunt).

Glutamate is the product of BCAA transamination, yielding the three ketoacids α-ketoisovalerate, α-ketoisocaproate, and α-keto-β-methylvalerate which, after conversion to succinyl-CoA or acetyl-CoA, enter the TCA cycle [75]. In actuality, BCAA metabolism in the brain is modest [65].

## 3. Metabolic Alterations in AD and PD Brains

Neurodegenerative processes are characterized by changes in the utilization of energy sources—mostly glucose—by the entire brain and its individual regions. Understanding how these alterations could be the cause and/or the result of neurodegenerative processes, and which mechanisms are involved in aberrant brain metabolism, is crucial for the development of treatments for neurodegenerative diseases [76].

### 3.1. Glucose Metabolism

Studies on glucose metabolism in AD brains are extensive (see Figure 1B). [^18^F]-fluorodeoxyglucose positron emission tomography (FDG-PET) studies led to the discovery of glucose hypometabolism AD brains. The regions known to be vulnerable to AD pathology—such as the hippocampus, lateral and medial temporal lobes, and posterior cingulate/precuneus—are the areas most severely affected by glucose hypometabolism [77,78,79]. This association can be attributed to the fact that the cerebral metabolic rate of glucose (CMRglc) is an important indicator of neuronal and synaptic activity and is correlated with the main clinical features of AD [77]. The reduction in the CMRglc evidently predicts the progression from mild cognitive impairment (MCI) to AD, with greater than 80% accuracy [80]. Additionally, conditions associated with a high risk for AD—such as carrying the ApoE4 allele [81,82] or being prediabetic/diabetic and elderly [83]—also show AD-like reductions in the CMRglc, even without any clinical manifestation of the pathology. Also associated with lower CMRglc is the progressive increase in glucose concentrations in the posterior cingulate/precuneus of AD brains [84], as measured with magnetic resonance imaging (MRI), since decreased glucose utilization leads inevitably to increased concentration of remaining intra- and/or extracellular glucose [84].

Studies using MRI and FDG-PET have shown that patients with PD have extensive areas of glucose hypometabolism [85,86], which correlates with impaired cognition [87], to the point that hypometabolism and atrophy represent stepwise stages of the neurodegenerative process in most of the cortical regions affected in Parkinson’s disease [88] (see Figure 1C).

As noted above, cerebral glucose uptake has been considered to be mainly independent of the action of insulin [12]. However, the expression of insulin-sensitive GLUT4 in the brain confirms that glucose metabolism in the brain is at least partially regulated by insulin [89]. The colocalization of GLUT3- and GLUT4-expressing cells further indicates that insulin plays an important role in regulating brain glucose [90]. In general, aging is characterized by chronic hyperinsulinemia that is associated with reduced expression of insulin receptors and impaired insulin signaling cascades (i.e., insulin resistance) [91,92]. Consequently, insulin desensitization in the brain increases the risk of developing neurodegenerative diseases, such as AD and PD. Analyses of the brain tissue of AD and PD patients show insulin desensitization, independent of a previous history of type 2 diabetes mellitus (T2DM) [93,94,95,96,97].

Chronic inflammation is one of the main drivers of growth factor desensitization—not just to insulin and IGF-1. Indeed, a variety of key neuronal growth factors, such as brain-derived neurotrophic factor (BDNF) and nerve growth factor (NGF), are downregulated by chronic inflammation [98,99]. Because of chronic inflammation, the activation of microglia leads them to secrete pro-inflammatory cytokines, such as TNF-α [100,101,102], which can, in turn, downregulate insulin signaling [103].

Since insulin in part regulates glucose uptake in neuronal and glial cells, alterations in the insulin cascades may be implicated in glucose hypometabolism associated with AD. In AD brains, insulin signaling is inhibited, and this inhibition is closely connected to inefficiency in glucose metabolism [104]. The impairment of insulin signaling is also involved in abnormalities in mitochondrial structure and function [105], as well as following activation of the mechanical target of rapamycin complex 1 (mTORC1) [9,106]. In addition, significant alterations in gene expression observed in the AD brain are connected to the generation and transmission of insulin signals [107]. The insulin-degrading enzyme (IDE)—which plays an important role in the degradation of amyloid beta (Aβ) monomers [108], and is reduced in AD brains [109]—is competitively inhibited by insulin, leading to the elevation of extracellular Aβ levels [110]. In vitro Aβ oligomers reduce plasma membrane insulin receptors and promote oxidative stress and synaptic spine deterioration [111].

In the 3xTg-AD murine model, oxidative stress, after inducing initial activation of IRS-1, activates negative feedback mechanisms to turn off IRS-1 hyperactivity, causing brain insulin resistance [20]. This alteration might contribute to impaired glucose metabolism, BBB dysfunction, and energy supply shortage, which are common hallmarks of diabetes mellitus, and can further contribute to Aβ generation [112,113]. Thus, insulin resistance might be a major cause of energy deficiency in AD brains, which is related to the manifestation of diabetes. In this respect, AD has been termed a neuroendocrine disorder, identified as “type 3 diabetes”, mirroring a new mechanism of neurodegeneration [20,96,114].

Brain mRNA levels of insulin receptors decline in age—especially in the hypothalamus, cortex, and hippocampus—and this is connected to chronic secondary hyperinsulinemia [92,115], which is enhanced in PD (Figure 1C). Studies found a significant loss of insulin receptor mRNA in the substantia nigra pars compacta (SNpc) of patients with PD with increased insulin resistance compared with age-matched controls [116,117,118]. Moreover, an increase in levels of IRS phosphorylation at serine residues that deactivates insulin signaling is also observed in the basal ganglia and substantia nigra [94]. Additionally, these alterations reportedly may precede the death of dopaminergic neurons [94]. IRS-1 Ser312 phosphorylation is increased in neurons in the putamina of PD patients, whereas higher levels of Ser616 phosphorylation are found in the hippocampi of PD patients compared with controls [119] (Figure 1C).

Phosphorylation of IRS-1 on serine residues is a critical component of intact insulin signaling, and prevents insulin/IGF-1 from binding to the IR and causing subsequent activation of downstream effectors. This is consistent with other studies that found elevated levels of IRS-1 pSer307, pSer312, and pSer616 connected with neuronal insulin resistance in AD [93,120] (Figure 1B). However, an important aspect that needs to be explored is whether brain insulin resistance is due to altered transit of insulin through the BBB, or whether the neurons themselves are no longer sensitive to the effects of insulin. Peripheral insulin, produced by the pancreas, crosses the BBB and exerts effects on the brain—especially on the hypothalamus. Studies have found that hypo- and hyperinsulinemia have little effect on total brain insulin, suggesting that brain insulin resistance may be due to decreased responsiveness to endogenous insulin [121].

Brain cells can metabolize glucose to ATP by either oxidative or non-oxidative metabolism. Whereas non-oxidative metabolism produces considerably less ATP than oxidative phosphorylation, it is preferentially used by the brain because it also supplies substrates for physiological processes such as synaptogenesis, myelination, and axonal elongation, producing less reactive oxygen species than oxidative phosphorylation [122,123].

While aerobic glycolysis gradually decreases with aging, oxidative glucose use remains unaffected. Consequently, reduction in aerobic glycolysis mainly contributes to the total reduction in glucose utilization, which is one of the main risk factors for AD [124].

Glucose metabolism is a well-orchestrated process that involves adjacent brain cells of different types. While astrocytes predominantly metabolize glucose via glycolysis, neurons depend on oxidative metabolism. Studies have shown that in aged mice the intercellular metabolic “crosstalk” between glia and neurons is disrupted [125]. This impairs the lactate shuttle, so neurons rely mainly on their own glycolysis and oxidation for energy, reducing their capacity for energy generation [125]. Cerebral ATP production is reduced by ~20% in early AD, and a further reduction occurs in the advanced stages of the disease [126]. In addition to reduced glucose metabolism, decreased ATP production can be attributed to an inadequate cellular uptake of glucose because of a reduced number of GLUTs observed in brain cells of different types [127,128].

Postmortem studies in individuals with AD have demonstrated a decrease in GLUT1 and GLUT3, especially in the parietal, frontal, temporal, and occipital cortices, the caudate nucleus, and the hippocampus [127,128,129]—coincidentally the same brain regions that demonstrate glucose hypometabolism in FDG-PET studies [127] (Figure 1B). Furthermore, the number of hyperphosphorylated tau-containing neurofibrillary tangles (NFTs) [128] and tau pathology [130] correlates with GLUT reductions in AD. In vitro, the Aβ peptide induces dysfunctional GLUT3, which leads to decreased glucose uptake despite increased GLUT3 expression [131]. The Aβ-induced reduction of GLUT1 and -3 function lowers protein O-GlcNAcylation, which is neuroprotective [132]. However, decreased GLUT3 function is associated with tau hyperphosphorylation [128].

In postmortem brain tissue from AD patients, GLUT2 overexpression was observed. Since GLUT2 overexpression is associated with increased expression of the astrocytic marker glial fibrillary acidic protein (GFAP), the increased GLUT2 expression in AD brains may be the result of astrocytic activation [128], which is likely explained as a mechanism to supply additional astrocytic GLUTs in order to shuttle energy to neurons though astrocytes [128] (Figure 1B).

In addition to changes in GLUT, alterations in glycolysis are observed in AD (Figure 1B). The relationship between glycolysis and aging in the brain [133] is complex. Glycolytic dysfunctions can cause age-related neurodegeneration [134], and in murine models of aging, the levels of glycolytic intermediates—such as G6P and F1,6BP—are inversely correlated with age [135]. However, astrocyte-specific glycolytic flux increases with age [136], and is associated with a decline in the resting cerebral blood flow [137]. Furthermore, reducing glycolytic flow via 2-deoxyglucose (2DG)—a known inhibitor of the glycolytic processes [138]—prevents neurodegeneration by reducing microglial inflammatory activity [133]. Additionally, 2DG reduces the effect of Aβ on neuronal cells [139]. Increasing NADH levels reverses the aforementioned effects of 2DG [140], and reducing the NADH/NAD^+^ ratio is suggested to be a possible way to attenuate AD-associated pathology [141]. In many studies, increased reliance on glycolysis and suppression of mitochondrial respiration confers increased neuronal resistance and survival [142]. From the above-mentioned studies, it can be concluded that upregulation of glycolysis in neurons may act as a compensatory mechanism against AD pathology. Although this compensation might be initially beneficial in AD, it will eventually be detrimental once the disease progresses to a stage in which the brain displays glucose hypometabolism—even in preclinical stages of AD [143].

Enzymes involved in glycolysis have also been evaluated, such as hexokinase (HK), glyceraldehyde 3-phosphate dehydrogenase (GAPDH), and pyruvate kinase (PK) [9,144,145] (Figure 1B). HK and PFK expression are lower in AD brains, and this is linked to dysregulated Wnt signaling, which is known to exert neuroprotective effects by promoting glucose metabolism [146]. However, HK activity can be competitively inhibited by G6P [147], which accumulates in AD [148]. GAPDH is involved in the sixth step of glycolysis, and catalyzes the conversion of glyceraldehyde 3-phosphate (GAD3P) to 1,3-bisphosphoglycerate, increasing the NADH/NAD^+^ ratio [149]. GAPDH can promote Aβ amyloidogenesis in vitro [150], and in the S-glutathionylated form might represent a blood marker of neuronal death during AD progression [151]. In the 5xFAD murine model of AD, GAPDH expression was increased by the inflammatory response generated by Aβ, which reportedly induced a shift from OXPHOS to glycolysis via the mTOR-HIF-1α pathway [23].

PK is a rate-limiting enzyme in glycolysis, and has four isomers: M1, M2, L, and R. PKM2, which regulates the levels of glycolytic intermediates along with ATP, is connected to neurodegenerative diseases [152] (Figure 1B). The upregulation of the Wnt/β-catenin pathway can promote glycolysis, which is related to PKM2. In AD, the Wnt/β-catenin pathway is downregulated through the partial inactivation of PKM2, and this is associated with oxidative stress and cell death [152].

As discussed further below, our laboratory used redox proteomics and enzyme activity assays to identify several glycolytic and TCA enzymes as oxidatively modified and dysfunctional in brains from subjects with AD and MCI, as well as animal models thereof [9,153,154,155].

Glycolytic dysfunctions have also been observed in PD (Figure 1C). Decreased glucose metabolism has been found to be associated with abnormally elevated levels of lactate and pyruvate in PD patients [156,157,158,159]. Interruption of glycolysis in astrocytes and oligodendrocytes leads to axonal damage and neurodegeneration [160,161].

Interestingly, α-synuclein (α-syn) aggregation is promoted by glucose deprivation [162]. Conversely, lactate reportedly exerts an opposing effect on α-syn [163]. Glycolysis is upregulated in response to mitochondrial dysfunction, and ATP generation via glycolysis has a protective role when complex I is inhibited [164,165,166,167,168,169]. Indeed, failure by neuronal cells to upregulate glycolysis seems to make them more sensitive to mitochondrial dysfunction [170].

Phosphoglycerate kinase (PGK) catalyzes the ATP-generating step of glycolysis, in which a phosphate group in 1,3-biphosphoglycerate is transferred to ADP, with the production of 3-phosphoglycerate and one molecule of ATP. Deficiency of PGK activity caused by genetic mutations (e.g., c.649G > A), which leads to defective ATP production, has been shown to be a major cause of medical conditions related to PD, such as neurological deficits, hemolytic anemia, and myopathy [171]. Multiple studies have shown that patients with a deficit of PGK activity exhibit PD-like symptoms, highlighting the role of PGK deficiency in the development of idiopathic PD [172,173,174,175]. These clinical findings have been further explored in studies conducted in preclinical models. Indeed, in a *Drosophila* model, PGK knockdown induced in dopaminergic (DA) neurons resulted in locomotive defects characterized by significant reductions in ATP and dopamine levels, with a stepwise loss of DA neurons [175]. Moreover, in different toxin-induced or genetic PD models, treatment with terazosin—a PGK agonist—enhanced brain ATP and dopamine levels with the restoration of motor function, suggesting that stimulation of PGK and glycolytic activities could represent a possible therapeutic approach in the treatment of PD [174].

Other studies found that PD-related genes—such as *PARK2* (Parkin), *SNCA* (α-synuclein), *PINK1*, and *PARK7* (DJ-1)—indirectly regulate glycolysis by interfering with different signaling proteins, including p53, HIF-1α, and AMPK [176,177,178,179].

Additionally, methylglyoxal (MGO)—a byproduct of the metabolism of GAD3P and dihydroxyacetone phosphate (DHAP)—is a potent glycation agent that quickly binds nucleic acids, lipids, and protein lysines/arginines to produce advanced glycation end products (AGEs), which have been reported in PD [180]. MGO induces mitochondrial dysfunction, and is detoxified by the glyoxalase system through the activity of glyoxalase-1 and -2 (GLO1–2), with GSH used as a cofactor [181,182,183]. Additionally, Parkin was recently shown to regulate glucose metabolism via ubiquitinylation of pyruvate kinase M1 and PKM2, which leads to a decrease in their enzymatic activity [184].

In PD brains, GAPDH, aldolase A, and enolase 1 are oxidatively modified by the lipid peroxidation product 4-hydroxynonenal (4-HNE) [185] (Figure 1C). These glycolytic enzymes are subjected to interaction and sequestration by amyloid-like structures such as α-syn fibrils [186,187]. GAPDH has been found to directly regulate α-syn aggregation and apoptotic neuronal cell death in an independent manner different from its role in glycolysis (Figure 1C).

Furthermore, GPI-1 was recently shown to have a protective effect against proteotoxic stress induced by α-syn in dopaminergic neurons, and this effect was demonstrated to be linked to glycolysis [188]. Notably, other studies have found a possible interaction between α-syn and glycolytic enzymes such as aldolase [189,190].

### 3.2. Ketone Bodies

As stated above, under normal physiological conditions, the brain primarily utilizes glucose for energy production. However, in situations where glucose is low, such as during prolonged fasting, ketone bodies become an important energy source for the brain. Indeed, infants who are breastfed utilize ketone bodies almost entirely. Neurodegenerative diseases are characterized by a deterioration of brain glucose metabolism, which naturally enhances ketone utilization.

Monocarboxylate transporters (MCTs) are a family of 14 receptors that are responsible for the passive transport of lactate, pyruvate, and ketone bodies into the brain [191]. During aging, the brain favors ketone metabolism by increasing neuronal MCT2 and decreasing astrocytic MCT4—perhaps as a compensatory response to glucose hypometabolism [192]. This hypothesis is further supported by the observation that the elevation of ketones in aged rat brains correlates with the rise of markers of mitochondrial dysfunction [193].

The main source of ketones for peripheral organs and the brain is the liver. However, myelin catabolism can be a source of ketones when peripherally produced ketones are not available to the brain—an age-related condition of reduced ketone transport [193] due to decreased MCT1 expression in the BBB [192], which may cause white matter degeneration [193]. In particular, the activation of the cytosolic phospholipase A2 (cPLA2)-sphingomyelinase pathway, induced by the age-induced decline in mitochondrial respiration and increased oxidative stress, leads to the catabolism of myelin lipids into fatty acids. Then, astrocytes can further catabolize fatty acids to produce ketone bodies that can enter the neurons via MCTs. This phenomenon may contribute to the demyelination observed in AD, and might play a pathogenic role [194]. Cerebral ketone uptake is proportional to peripheral levels of ketone bodies, and this ability is conserved in the AD brain, underlining that in AD brains normal ketone metabolism occurs [195]. Notably, PET studies in MCI and AD individuals showed that brain regions characterized by glucose hypometabolism metabolize acetoacetate normally [78,196]. In the 3xTgAD murine model, hippocampal MCTs were altered [192]. In AD, although glial MCT1 and MCT4 protein expression decreases, neuronal MCT2 protein expression increases [192]. These alterations occur concomitantly with reductions in GLUT1 and GLUT3 protein expression [192], highlighting a compensatory mechanism executed to deal with glucose hypometabolism. This observation is substantiated by studies in female 3xTgAD mice, in which the ketone-metabolizing succinyl-CoA:3-ketoacid coenzyme A transferase (SCOT) enzyme is inversely correlated with pyruvate dehydrogenase (PDH) [197]. These changes in enzymatic activity indicate a bioenergetic shift from glucose toward ketones as metabolic substrates in AD [197].

The ability of the brain to utilize ketones in other neurodegenerative diseases, such as PD, has not been studied in depth. However, it has been hypothesized that, like AD, the pathology of PD may be associated with preserved metabolism of ketones, which could offset the energy deficit due to glucose hypometabolism. This is confirmed by the positive effects of ketogenic interventions in experimental models of PD [198].

### 3.3. TCA Cycle and OXPHOS

Cerebral glucose hypometabolism—characterized by reduced glucose uptake and utilization associated with brain insulin resistance [20,96,114,199]—and progressive mitochondrial dysfunction with aging [200,201] have recently been correlated with AD, and suggest the involvement of energy metabolism alterations in AD’s pathophysiology (Figure 2B).

Mitochondrial energy production involves electron transfer between the enzymes of the TCA cycle, generating the reducing coenzymes NADH and FADH_2_, and successive oxidation of these factors on the complexes of the ETC. At the end of glycolysis, pyruvate is oxidized to acetyl-CoA in order to access the TCA cycle, or is reduced to lactate, depending on the redox status of oxidized/reduced forms of NAD^+^. In AD transgenic mice, the NADH pool is reduced with age, and the redox state becomes more oxidized [202,203,204,205].

Proteomic analysis revealed that in aging, alterations of NADH levels correlate with dysfunction of TCA enzymes, such as upregulation of fumarase 1 (FH1), malate dehydrogenase 1 (MDH1), SDH, PDH, and subunits of complex I [206]. Reduced TCA cycle metabolism is also correlated with the downregulation of isocitrate dehydrogenase 1/2 (IDH 1/2) and a subunit of succinyl-CoA synthetase in aged murine brains. In particular, the downregulation of IDH leads to lower NADPH and 2-OG, known for their protective role against oxidative stress, resulting in inefficient ROS clearance [207,208]. AD brains display reductions in IDH, 2-OGDH, and PDH complexes [209], although the activities of MDH and SDH are increased [210]. Citrate synthase (CS) activity appears to be negatively regulated by ApoE4 [211] and decreased in AD patients [212]. As a downstream product of 2-OG, succinyl-CoA may also be reduced [133] (Figure 2B).

TCA cycle intermediates—such as citrate, cis-aconitate, and 2-OG—are altered in the plasma and cerebrospinal fluid (CSF) of patients with AD or MCI [213,214]. This is mirrored by reduced levels of ketogenic and glucogenic amino acids, which produce intermediates that flow into the TCA cycle. In contrast, high concentrations of creatinine are observed in the CSF of AD patients, indicative of a disrupted creatine–phosphocreatine shuttle [215] under conditions of inadequate glucose intake. Therefore, lactate, amino acids, and fatty acids represent an alternative energy source for neurons during hypoglycemia [52], with lactate acting as a neuroprotective metabolite [216] via transcriptional activation of brain-derived neurotrophic factor expression [217].

In both aging and AD mouse models, the flux to the TCA cycle is increased together with the levels of acetyl-CoA and NADH, which are positively correlated with age and AD progression [218]. Glutamine and fatty acid metabolism are upregulated with age and AD, possibly indicating a cellular requirement for additional energy production [218]. In the *APP/PS1* mice with early-stage disease, the levels of succinic acid, 2-OG, citric acid, cis-aconitic acid, and fumaric acid are decreased [219,220]. In the same model, ^13^C flux analysis detected reduction in energy metabolism as well as neurotransmitter precursors (glutamine, glutamate, γ-aminobutyric acid, and aspartate) [221], and this observation was correlated with accumulated brain glucose. This result indicates that decreased glucose responsiveness in AD could induce compensatory activation of alternative sources—both glucogenic and ketogenic—to fuel the TCA cycle, such as fatty acids and amino acids [222]. In support of this hypothesis, plasma from AD patients displays elevated carnitine forms of major fatty acid oxidation intermediates such as acetyl-carnitine (C2) and long-chain acyl-carnitines (LCACs, from C6 to C18) [213].

In addition to AD, alterations in glucose metabolism have also been reported to occur in PD, such as a decrement in glucose flux and atypically high levels of lactate/pyruvate [156,157,158] (Figure 2C). Furthermore, dysregulation in the TCA cycle has been reported in PD brains [156,223]. Not only mitochondrial energy dysfunction, but also genetic variations are related to mitochondrial changes in early-onset PD. As mentioned above, alterations in the *PINK1*, *Parkin*, *SNCA*, and *DJ-1* genes alter mitochondrial morphology [224,225,226,227]. α-Syn enters the mitochondria in a manner dependent on energy state [228], and once accumulated in the mitochondria causes complex I dysfunction, increased ROS production, and reduced ΔΨm, which exacerbate the mitochondrial injury present in old substantia nigra neurons in PD [229,230] (Figure 2C).

Disturbances in OXPHOS involve many effects on cellular homeostasis, e.g., (1) promoting the accumulation of NADH and FADH_2_ in mitochondria, (2) reductions in ATP production, and (3) increased ROS production. The brain is an organ with high oxygen consumption and low antioxidant defenses; thus, the brain is vulnerable to oxidative stress [133].

The elevation of NADH and ROS levels in mitochondria inhibits the activity of TCA enzymes, leading to accumulation of the TCA intermediates [231]. Changes in the 2-OG/succinate levels modify the activity of NAD^+^-dependent 2-OG-oxygenases (2-OGDO)—enzymes controlling the epigenetic modifications of chromatin [232]—ultimately perturbing neuronal function. The impaired OXPHOS in AD could induce the accumulation of citrate and 2-OG, which are two potent epigenetic regulators [233,234]. In particular, 2-OG can induce random changes in DNA and histone methylation, leading to an epigenetic drift in gene expression, such as in the aging process and AD [235].

Complex I protein levels are significantly reduced in the temporal, parietal, and occipital cortices in AD brains [236]. The biosynthesis of the 24 KDa subunit of complex I is lower in the temporal and occipital cortices, while the 75 KDa subunit complex I is lower in the parietal cortex of AD brains [237]. Complex III protein levels are reduced in the temporal cortex [238], whereas complex V proteins are reduced in the hippocampus of AD brains [239]. The activity of cytochrome c oxidase (complex IV) is altered in the brain areas affected by AD [240,241,242], in a region-specific fashion; reportedly, it is lower in the temporal, frontal, and parietal cortices, but higher in the hippocampus [243] (see Figure 2B). A decreased expression of subunit 4 in the cytochrome c oxidase complex was noted in transgenic AD mice [244]. Furthermore, Aβ can induce ROS production in neuronal mitochondria, disturbing complex IV functions [245]. The Aβ fragment 25–35 reduced the activity of complex IV without changing the activity of the other respiratory complexes in isolated rat brain mitochondria [246]. Two caveats of studies employing Aβ25-35 are (a) the mechanism of ROS production with a terminal Met residue is different from that of Aβ1-42 with an interchain Met-35 residue [247]; and (b) There is no reported evidence of Aβ25-35 in AD brains, making studies using this Aβ peptide fragment of academic interest, but not of relevance to AD.

Mitochondria are not only the principal source of ROS, but also an important target of ROS attack, leading to a vicious cycle in which oxidative stress can further exacerbate mitochondrial dysfunction [6,248]. As mentioned above, the activity of the ETC complexes is considerably reduced in AD, leading to compromised OXPHOS [249]. This phenomenon has been established in mitochondria isolated from 3-month-old AD mice [250] and brain tissue from AD patients [239]. The damage to mitochondrial respiratory function in AD patients may be caused by the effect of Aβ on mitochondrial OXPHOS capability, and may also be correlated with Aβ levels [251] (Figure 2B).

The above-mentioned lower TCA cycle flux in AD might suggest that the ETC-OXPHOS is deprived of NADH for ATP generation [210]. The NAD^+^/NADH redox couple is a potential sensor for dehydrogenases, and acts as a switch to affect the rate or direction of the cellular metabolic flux. Moreover, as discussed above, in AD, metabolic upregulation of fatty acid β-oxidation is exploited to generate NADH as an alternative to carbohydrate oxidation, so as to maintain redox balance and maximize energetic function [218].

Mitochondrial dysfunction—specifically a deficiency in complex I of the ETC—is prominent in PD [252], although its deficiency seems to be limited to regions of the brain that are pathologically altered in PD [253,254] (Figure 2C). However, mitochondrial complex I deficiency and oxidative stress appear to be key factors in PD’s pathogenesis [255]. These are interconnected, as inhibition of complex I results in increased production of ROS which, in turn, inhibit complex I. Over time, this vicious cycle in dopaminergic neurons leads to excessive oxidative damage and ATP deficiency that will eventually lead to cell death [256,257,258,259]. Evidence supporting the energy failure of PD brains includes the creatine kinase (CK)-mediated increase in ADP phosphorylation at the expense of phosphocreatine, which is linked to upregulated creatine synthesis at the expense of amino acids such as glycine [252] (Figure 2C). These mechanisms underscore energy inefficiency and mitochondrial dysfunction in PD [260]. As noted above, alternative energy sources have been shown to protect against PD neurodegeneration; most studies show that glycolysis is upregulated in response to mitochondrial dysfunction, and ATP generation via glycolysis plays a protective role against complex I inactivation [164,167,169].

### 3.4. Aminoacid Metabolism

Several findings support the notion that glucose metabolism, mitochondrial dysfunction, and metabolism of BCAAs are altered in the brains of AD models [261] (Figure 3B). A significant reduction in valine found in AD CSF [262] has been recently confirmed in newly diagnosed AD patients [263]. Lower plasma valine levels were correlated with the rate of cognitive decline [264]. Reduced levels of BCAAs in the blood were found to be associated with an increased risk of dementia and AD [265]. The decreased levels of BCAAs could affect glutamate synthesis, thereby impairing neurotransmission. Indeed, in line with lower BCAA levels in AD, a reduction in glutamate levels was reported in AD patients [266], together with decreased levels of glutamine [267]. Furthermore, since glutamate, as an excitatory neurotransmitter, binds to cell surface receptors such as α-amino-3-hydroxy-5-methyl-4-isoxazolopropionic acid (AMPA) receptors and N-methyl-D-aspartate (NMDA) receptors [268], and since reduction in NMDA receptor function relates to Ca^2+^ dysregulation and reduced synaptic plasticity [269], it is conceivable that reduced BCAA levels contribute to dementia in AD [265] (Figure 3B).

The enzyme GS plays a key role in brain function. In normal astrocytes this protein, by catalyzing the rapid amidation of glutamate, forms glutamine, and in this way contributes to establishing the correct levels of glutamate and ammonia and, consequently, to modulating the excitotoxicity that results from impairment of the glutamate–glutamine cycle. In AD brains, the conversion of glutamate to glutamine by GS occurs less efficiently than in control brains [270,271,272]. Moreover, both the glutamate transporter GLT1 [273] and GS [274] are oxidatively modified and dysfunctional in AD [254,259], potentially exposing neurons to glutamate excitotoxicity that is extensive in AD brains [275] (Figure 3B). Protein oxidation might be part of the mechanism of neurodegeneration in AD brains [9,274,276,277,278,279,280,281].

Glutamate excitotoxicity is also extensive in PD brains, and this seems to be related to GLT1 downregulation [282] (Figure 3C).

Microglial metabolism plays a significant role in inflammatory responses during AD-associated neurodegeneration [283]. It is important to highlight the fact that GS activity in the microglia mitigates microglial inflammatory response, suggesting a novel mechanism by which GS loss of function amplifies inflammatory activity, leading to neurodegeneration [284]. Moreover, GS inhibition reduces insulin-related glucose uptake in the microglia, suggesting GS activity as a potential unifying mechanism controlling insulin resistance, inflammation, and metabolism [284].

Accumulation of N-acetyl aspartate (NAA), together with alterations of metabolites such as aspartate, glutamate, citrate, malate, pyruvate, serine, and lactate, are found in the frontal cortex samples of AD subjects. This suggests that the amino acid transport mechanism between mitochondria and the cytosol could be compromised in AD brains [285]. During early postnatal CNS development, NAA production in neurons is increased. NAA is transported from neurons to the cytoplasm of oligodendrocytes, where aspartoacylase (ASPA) cleaves the acetate moiety to promote synthesis of fatty acids and steroids as building blocks for myelin lipid synthesis [286] (Figure 3B). Previous findings indicate that cholinergic treatment could induce elevated NAA levels in AD [287,288], and that this effect could be reversed by other therapeutic strategies [289,290], suggesting a possible influence of dietary regimens or pharmacological treatments on NAA levels measured in AD subjects. The recent finding that NAA could mitigate the inflammatory response of macrophages through NMDAR interaction [291] clearly opens interesting clues with respect to the role of this metabolite in the neurodegenerative processes underlying the pathologies of AD and PD.

Metabolomic studies have identified increased alanine and phenylalanine [292] and reduced tryptophan in PD brains [293]. Dysregulation of the kynurenine pathway—a metabolite derived from tryptophan—was found in PD [294,295,296], providing potential novel biomarker candidates for investigating the pathogenesis of PD and suggesting new therapeutic strategies [294]. This study confirmed the pioneering research of M. Flint Beal on decreased kynurenine in neurodegenerative disorders [297]. Alterations in phenylalanine—an initial metabolite in the biosynthesis of dopamine—may be corrected by treatment with different dopaminergic drugs [298]. In addition, L-DOPA treatment has a profound impact on aromatic amino acid metabolic pathways.

Metabolomics studies profiling the blood metabolomes of PD patients showed a significant increase in BCAAs in this disorder [299] (Figure 3C). In normal brain function, threonine and glycine can be converted to creatine, providing phosphate groups for ADP to produce ATP [252]. During the initiation of neurodegenerative processes associated with PD, the metabolism of glycine, serine, and threonine is downregulated [252], consistent with mitochondrial dysfunction in PD [260].

Proline is involved in aging and neurodegeneration [300,301]. In this regard, a recent study found high concentration levels of ornithine—the precursor of proline—in the sera of patients with PD [302], confirming results obtained by Çelik et al. [303]. Ornithine accumulation is associated with hyperosmolarity in different regions of the brain via urea cycle flux [303]. Additionally, ornithine accumulation leads to higher proline levels, which could induce collagen biosynthesis, leading to a shift in the immune system towards a program of wound healing [303]. Increased levels of trans-4-hydroxyproline were found in the CSF and sera of patients with PD—possibly partially caused by the intensified degradation of collagen [302,304].

### 3.5. Redox Status

The brain is characterized by high oxygen consumption (20% of the total bodily consumption is employed by the brain), high energy demand, and relatively low levels of antioxidant systems [305]. It follows that a common element in AD and PD is the presence of high levels of ROS, which are correlated with neuron death [306]. ROS are highly reactive, oxidizing, small molecules, in the form of hydrogen peroxide, superoxide radical anions, and other free radicals (such as hydroxyl radicals). The main endogenous sources of ROS are the respiratory chain in mitochondria, peroxisomal activity, NADPH oxidases (NOX), the endoplasmic reticulum, and activated inflammatory cells [307]. In the brain there are further ROS sources, such as Ca^2+^ signaling, metals, and neurotransmitters (Figure 4). ROS have many physiologically regulated functions in the brain. Indeed, microglia and astrocytes produce ROS with the purpose of regulating neurons and glial exchanges and neuronal activity [308]; this phenomenon is reported as “redox signaling” [305]. ROS can contribute to the activation of guanylate cyclase, leading to the production of cGMP—an important second messenger. Additionally, ROS are involved in the activation of the transcription factor nuclear factor κB (NF-kB) [306]. The superoxide anion and hydrogen peroxide originating from NOX2 act on the PI3K/Akt pathway, with a beneficial effect in maintaining stem cell proliferation in the hippocampus [309], potentiating learning and memory [310]. Interestingly, NOX2 deficiency leads to cognitive loss [309]. Then, the hippocampus is highly affected in AD [311]. Moreover, NOX-derived H_2_O_2_ plays a role in axon development [312], managing the correct innervation [313]. Thus, ROS contribute to the potentiation of synaptic plasticity, neuronal development, and polarization [314] in physiological conditions (Figure 4).

However, ROS can be highly reactive and, therefore, dangerous. In response, cells have developed many enzymatic (superoxide dismutase (SOD), catalase (CAT), glutathione peroxidase (GPx), thioredoxin (TRX), peroxiredoxin (PRX), glutathione reductase (GR), glutathione S-transferases (GST)) and non-enzymatic systems (glutathione (GSH), vitamins A, C, E, b-carotene) to keep ROS production under control [315]. Among these, glutathione represents the most abundant endogenous antioxidant molecule in the brain [316]. GSH is synthesized from three amino acids: glutamate, glycine, and cysteine (Cys), with Cys being the most limiting reagent compared to the others, to the point that its shortage could restrain GSH de novo synthesis [317]. A GSH de novo synthesis precursor, N-acetyl-cysteine, was shown to be useful in improving memory deficits in AD patients [318,319] and, when given in drinking water to a pertinent AD murine model, to significantly decrease oxidative damage in the brain [320]. Glutathione is characterized by a reactive thiol group (GSH) that can combine with free radicals, becoming oxidized (GSSG). GPx is mostly responsible for this reaction, which also can occur spontaneously. GR then reduces GSSG back to GSH, using NADPH for reducing equivalents. Beyond free radicals, GSH can also react with other nucleophilic molecules, due to the action of GST, forming glutathione-S-conjugates—a means of detoxifying toxic compounds—and GST is defective in AD brains [321]. GSH is also involved in the glutathionylation of proteins—a mechanism protecting proteins from oxidation. It was reported that in AD many proteins are glutathionylated [322,323].

Among the alterations observed in AD brain metabolism, the PPP’s impairment impacts on redox balance, because it can generate NADPH equivalents which, as noted above, are useful to reduce GSSG derived from ROS back to GSH [133]. The rate-limiting enzyme G6PD has been shown to exert a role in neuroprotection [324]. Moreover, G6PD activity was found to be decreased in aged murine brain cortices [325], as we;; as in the hippocampi of human AD brains [326], but also upregulated in AD [327]. Notably, Aβ peptides, tau aggregation, and ApoE in AD are involved in PPP impairment [328,329], as well as with the alteration of some metabolites, such as G6P, gluconolactone, and gluconate [330]. Additionally, Aβ damage can be hindered via deviation of glucose through the PPP [331,332]. In AD, increased levels of lactate together with reduced levels of R5P are indicators of PPP upregulation [333]. Furthermore, the activity of the other PPP rate-limiting enzyme, transketolase, has been shown to be lowered in AD [334,335].

Derangements of the PPP are accompanied by GSH abnormalities. With aging and in many neurodegenerative diseases, GSH levels are reduced [336]. In AD, the ratio GSH/GSSG is unbalanced toward the oxidized form [337], and GSSG levels are correlated with the decreased cognitive functions in AD patients [338]. Moreover, in AD patients, activities of GPx and SOD reportedly are low [339], with consequent H_2_O_2_ accumulation. In such conditions, ROS can damage macromolecules and structures such as membranes, proteins, lipids, and DNA [340] (Figure 4).

Oxidative stress also involves cell metabolism and signaling [9,114], and often triggers a pro-inflammatory response, with increased cytokine production, observed in AD and PD [341]. In this state of chronic inflammation, astrocytes can release pro-inflammatory factors together with free radicals, leading to microglial activation [342]. Astrocytes also supply GSH to other brain cells, but the inflammatory activation associated with neurodegeneration reduces the intracellular GSH levels through p38 MAP-kinase, Jun-N-terminal kinase, and NF-κB in human microglia and astrocytes [343]. All of these events are often present in the early stages of the pathologies, and contribute to the shortage of the antioxidant defenses, favoring disease progression [344] (Figure 4).

Oxidative stress targeting proteins could have serious consequences, especially because the associated enzymatic activity can be impaired by oxidation [155,345]. Indeed, many enzymes have been reported to display lowered activity consequent to their oxidation in neurodegenerative diseases, including GAPDH, a-enolase [274,346], and GS [272,347]—with obvious consequences for brain function. The oxidation-dependent inactivation of the glycolytic enzymes leads to the alterations in glucose metabolism observed in AD [126,145] and, ultimately, to neurodegeneration [348,349].

The brain also is vulnerable to oxidative damage, due to its highly polyunsaturated-fatty-acid-rich structure which, because of its labile allylic H atoms, represents a target for lipid peroxidation. This event, in addition to producing damage to lipid structures, leads to the generation of neurotoxic aldehydes such as 4-HNE [280,350] or dienals, which can induce apoptosis [351]. Moreover, the brain uses high levels of Fe^2+^, which could play a role in oxidative stress [352] and could induce autophagy [353]. Moreover, Ca^2+^ homeostasis, which is important in signal transmission and action potential generation in neurons, could have impact on oxidative stress, and vice versa [9,354]. Ca^2+^ induces neuronal nitric oxide synthetase (nNOS), thus leading to NO synthesis [355], and NO is a free radical that can combine with superoxide free radicals to form peroxynitrite, which can lead to the nitration of key protein tyrosine residues, thereby interfering in important tyrosine phosphorylation-based intracellular signaling that is damaging to cells [356,357,358,359,360,361]. Nitration of heat shock protein 90 induces apoptosis in neurodegenerative diseases [362]. The interplay between Ca^2+^ and ROS also involves the regulation of mitochondria-associated membrane (MAM) formation [363]. These structures regulate many mitochondrial functions, and their dysregulation causes oxidative stress, associated with neurodegeneration in AD and PD [364]. Moreover, Ca^2+^ can regulate apoptosis through mitochondrial permeability transition pore (mPTP) opening [365]—a common mechanism in neurodegenerative disorders [305,366,367]. AD brains are also more vulnerable to oxidative stress, because of further decreased levels of antioxidant enzymes and small antioxidant molecules [368,369].

Finally, synaptic transmissions could trigger oxidative stress. Glutamatergic transmission—involved in PD as excitotoxicity—leads to depletion of GSH, because glutamate inhibits the cysteine transporter [370], which is essential for GSH synthesis [371]. Beyond GSH depletion, glutamate excitotoxicity is linked to Ca^2+^/ROS interplay and, consequently, could lead to apoptosis [305]. Glutamatergic transmission, through NMDAR, is also linked to protective mechanisms involving NOX2 [372].

Amine neurotransmitters—such as dopamine, noradrenaline, tyramine, and others—are metabolized by monoamine oxidases (MAOs)—mitochondrial enzymes that deaminate their substrates in the presence of O_2_, producing the related aldehyde, H_2_O_2_, and NH_3_. MAO catalysis requires flavin as a cofactor [373]. MAOs are present in two isoforms: MAO-A and MAO-B, which differ based on substrate specificity [374,375] and affinity for O_2_ [376]—an important feature, since their activities are influenced by oxygen’s availability. Both isoforms can generate peroxide at high rates in the brain under conditions of saturated O_2_. MAO-B is located in the mitochondrial intermembrane space, where the GPx4 isoform is also present [377] and counteracts H_2_O_2_ generation. Through H_2_O_2_ production, MAOs can also induce apoptosis, via a Ca^2+^-dependent mechanism [378]. Indeed, in AD and PD, MAO activity is enhanced [379,380], and MAO inhibitors have been proposed as therapeutic agents for AD [381] and are currently a therapeutic option for PD [382]. MAOs can also keep ROS generation under control, preventing neurotransmitter oxidation; in fact, while the MAO-catalyzed reaction produces a single hydrogen peroxide molecule, the oxidation of amine neurotransmitters would generate peroxide at high rates [383]. The MAO reaction is coupled with aldehyde dehydrogenase (ADH), which converts the aldehydes to the corresponding acids, detoxifying them. The inhibition of ADH has been related to PD [384].

As mentioned above, dopamine, along with adrenaline and serotonin, can auto-oxidize, generating ROS and quinones [385,386]. In particular, it has been observed that some metabolites of dopamine oxidation, such as 6-hydroxydopamine, play a role in PD [387], e.g., through increased mitochondrial ROS and inhibition of glucocerebrosidase—a lysosomal enzyme involved in the pathogenesis of PD [388,389].

At the metabolic level, all of the cell types in the brain (i.e., neurons, glia, astrocytes, endothelial cells), with all their differences, cooperate in the correct function of the whole tissue, thus performing a “metabolic coupling” [390]. Neurons obtain their energy mainly from oxidative phosphorylation (OXPHOS), whereas astrocytes obtain most of their energy from glycolysis, and the lactate generated can be used by neurons as an energy source. This cooperation also involves antioxidant defense [391]; astrocytes, when higher levels of mitochondrial ROS are produced, play a pivotal role in the antioxidant protection of neurons [392]. Indeed, the higher mitochondrial ROS production in astrocytes can induce the formation of nuclear factor erythroid 2-related factor 2 (Nrf2)—a transcription factor controlling the basal and induced expression of an array of constitutively active antioxidant responses [393], thus reducing the release of ROS by suppressing NOX1 and NOX2 expression. The induction of Nrf2 also contributes to high extracellular GSH levels, further contributing to the maintenance of the proper redox balance in neurons [394,395].

Some metabolism-related therapeutic treatments for Alzheimer’s disease and Parkinson’s disease are listed in Table 2.

## 4. Conclusions

The content of this review article indicates that cerebral energy metabolism involving glucose, ketone body, and amino acid metabolism is dysfunctional in the brains of persons with AD and PD—conditions that in AD happen early in the progress of the disease, well before dementia presents in patients [9,403]. In each type of brain metabolism, the involvement of mitochondria and oxidative and/or nitrosative stress iscritical to these altered metabolic processes in both AD and PD.

Because so many different aspects of brain metabolism are altered in AD and PD, we predict that targeting a single metabolic process will be insufficient to curb the progression of both disorders. Rather, targeting the common processes associated with both AD and PD—i.e., mitochondrial alterations and oxidative and/or nitrosative stress—may be a more promising therapeutic approach [248]. Brain-permeable, cell-membrane-passable, mitochondrially targeted antioxidant agents, such as Mn(III) *meso*-tetrakis(*N*-*n*-butoxyethyl-pyridinium-2yl)porphyrin, MnTnBuOE-2-PyP^5+^—also known as BMX-001 or MnP (an MnSOD mimetic) [404]—and mito-Tempol (a mitochondrially directed antioxidant), have shown promise in attacking certain ROS-associated cancers and other disorders [405]. It will be interesting to see whether future studies with these agents in AD and PD models, leading ultimately into clinical application for both disorders, prove to have disease-modifying properties. Attention would need to be paid to the baseline level of antioxidant potential, as well as the baseline level of oxidative stress already present in each participant in the clinical trials, which could otherwise confound any conclusions reached.

Since both Ab42 oligomers and a-Syn oligomers are associated with oxidative damage to mitochondria [7,9,248,406,407,408], in addition to the agents listed in Table 2, therapeutic molecules designed to specifically block both oligomeric types offer the strong possibility to halt—or at least slow—the progression of both disorders in the brain; future studies will determine whether this prediction is validated.

## Figures and Tables

**Figure 1 molecules-27-00951-f001:**
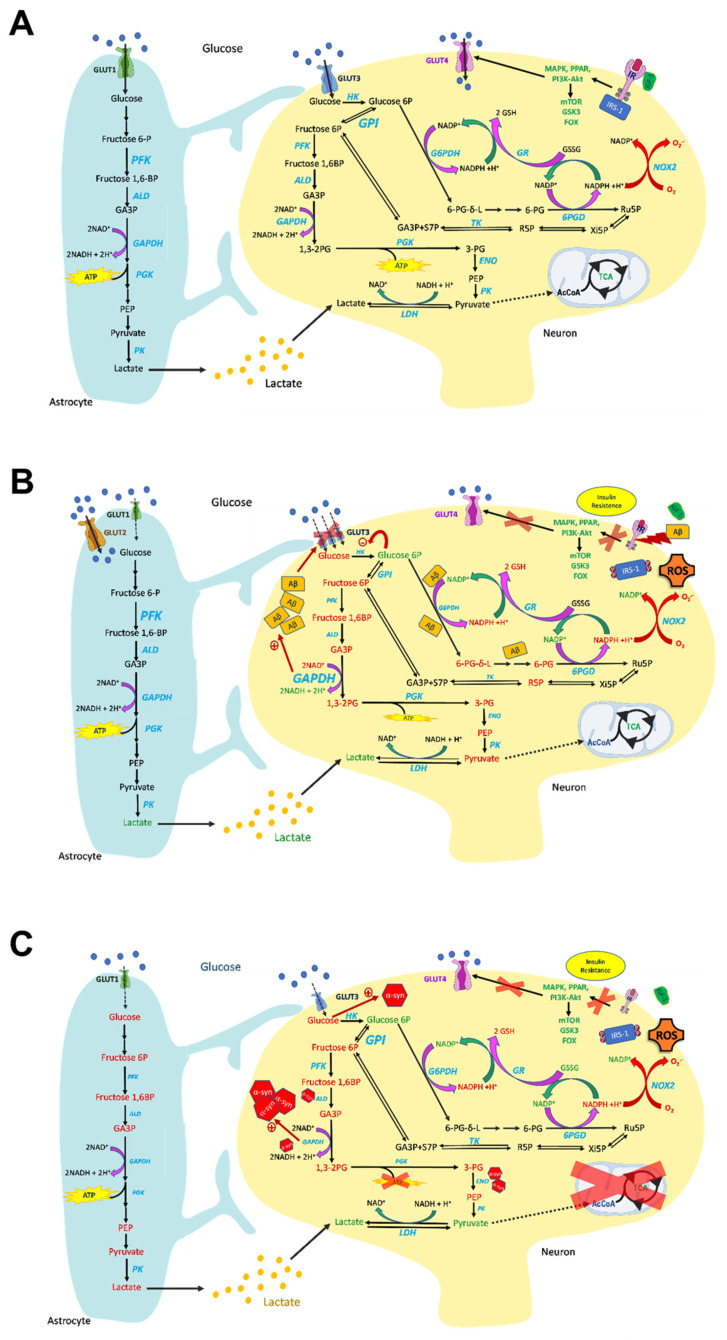
**Overview of glucose metabolism in normal (A), AD (B), and PD (C) brains.** (**A**) Glucose is the main energy substrate in the brain, as it is the principal source of ATP. In the brain, glucose is taken up from the bloodstream by specific glucose transporters, to be metabolized through glycolysis. Pyruvate enters the TCA cycle coupled with OXPHOS and ATP synthesis. Glucose transport in the brain occurs via GLUT1 in astrocytes and GLUT3 in neurons. Insulin-dependent GLUT4 transport also occurs in the brain, and its activation leads to insulin receptor signaling, occurring through the PI3K/Akt and MAPK pathways, regulating the brain’s main cellular functions. Glucose metabolism is prominent in the brain—especially in astrocytes—and strongly interconnected among the different cell types. Glucose 6-phosphate (G6P) can be channeled into the pentose phosphate pathway (PPP) to support NADPH synthesis, which is necessary to sustain the brain’s antioxidant defense, and this is enhanced by the recycling of fructose 6-phosphate (F6P) originating from the PPP back to G6P due to high GPI activity. (**B**) Glucose metabolism is impaired in AD brains. Signs of impaired insulin signaling cascade are present in AD brains, with insulin resistance and downregulation of insulin receptors, which contribute to brain glucose hypometabolism. In AD, decreased glucose metabolism impacts the metabolic crosstalk between astrocytes and neurons, as the lactate shuttle is impaired, leading to reduced ATP synthesis. GLUT1 and -3 are decreased in AD brains, and this correlates to glucose hypometabolism, and is a major pathological sign of AD, whereas GLUT2 increases, indicating prominent astroglial activation in AD brains. Glycolysis increases in astrocytes and microglia, and this is associated with neurodegeneration. G6P and fructose 1,6-bisphosphate (F1,6BP) levels are inversely correlated with age. Hexokinase (HK), PK, and PFK are downregulated in neurons, whereas GAPDH is upregulated and can promote Aβ amyloidogenesis. Aβ plays a role in impairing the PPP, leading to G6P accumulation—which inhibits HK activity—and to decreased defense against ROS. Aβ oligomers also reduce the IR and promote synaptic spine deterioration. (**C**) Glucose metabolism is dysfunctional in PD brains, and this mirrors a significant loss of IR. Furthermore, IRS phosphorylation deactivates insulin signaling, leading to insulin resistance. Moreover, the glucose transporters GLUT1 and GLUT3 are downregulated. The decrease in glucose metabolism, prominent both in neurons and in astrocytes, is associated with pyruvate and lactate accumulation and deleterious ATP depletion. Depletion of the ATP-generating enzyme PGK is associated with neuronal deficits with PD-like symptoms. Low glucose promotes α-synuclein (α-syn) aggregation. α-Syn fibrils interact with GAPDH, aldolase (ALD), and enolase (ENO), and their activities are consequently decreased. Furthermore, GAPDH directly regulates α-syn aggregation and apoptotic neuronal cell death. Increased metabolite levels are reported in green, whereas decreased levels are reported in red.

**Figure 2 molecules-27-00951-f002:**
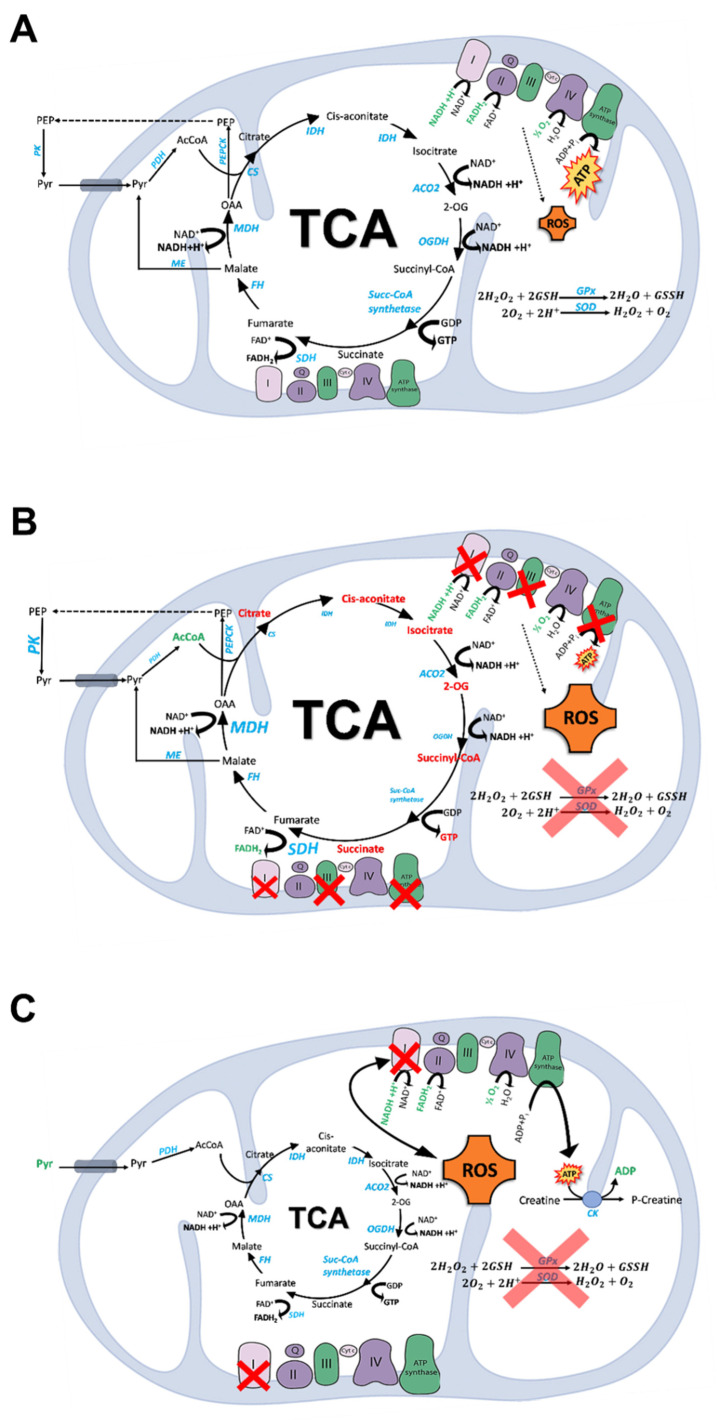
**Overview of the TCA and OXPHOS functions in normal (A), AD (B), and PD (C) brains.** (**A**) The TCA cycle oxidizes acetyl-CoA to two molecules of CO_2_, leading to the production of ATP and the reduction of NAD^+^ and FAD^+^ to NADH and FADH_2_, which enter the electron transport chain (ETC) complexes I and II, respectively. The electron flux from complexes I and II through the ETC leads to production of ATP by means of OXPHOS, which is coupled with the TCA cycle as it reoxidizes the coenzymes necessary for TCA function. ROS produced by OXPHOS are counteracted by Cu,Zn-superoxide dismutase, Mn-superoxide dismutase, peroxiredoxin, and glutathione. In the brain, pyruvate is recycled from malate and oxaloacetate through malic enzyme and phosphoenolpyruvate carboxykinase (PEPCK), respectively. (**B**) Aging is characterized by dysfunctions of the TCA cycle. In both aging and AD murine models, the levels of acetyl-CoA and NADH were increased, whereas the levels of succinic acid, 2-OG, citric acid, cis-aconitic acid, fumaric acid are decreased. This mirrors the reduction of IDH, 2-OGDH, PDH complexes, and CS with the increase in MDH and SDH in AD brains. Complex I, complex III, and complex V proteins are reduced in different regions of AD brains, leading to compromised OXPHOS. The activity of cytochrome c oxidase (complex IV) is lower in many brain regions, but is increased in the hippocampus. The damage to mitochondrial respiratory function in AD patients is associated with ROS formation, enhanced by the cellular inability to cope with the oxidative surge due to a lower antioxidant defense. (**C**) PD is associated with mitochondrial dysfunctions. In PD brains the lactate/pyruvate ratio is high, and this leads to TCA dysregulation. Mitochondrial complex I deficiency and oxidative stress are key factors in PD’s pathogenesis, and they are interconnected in a vicious cycle, in which a weakened antioxidant defense plays a role. Energy failure of PD brains leads to a creatine kinase (CK)-mediated increase in ADP phosphorylation at the expense of phosphocreatine, which is linked to upregulated creatine synthesis. Increased metabolite levels are shown in green, whereas decreased levels are shown in red.

**Figure 3 molecules-27-00951-f003:**
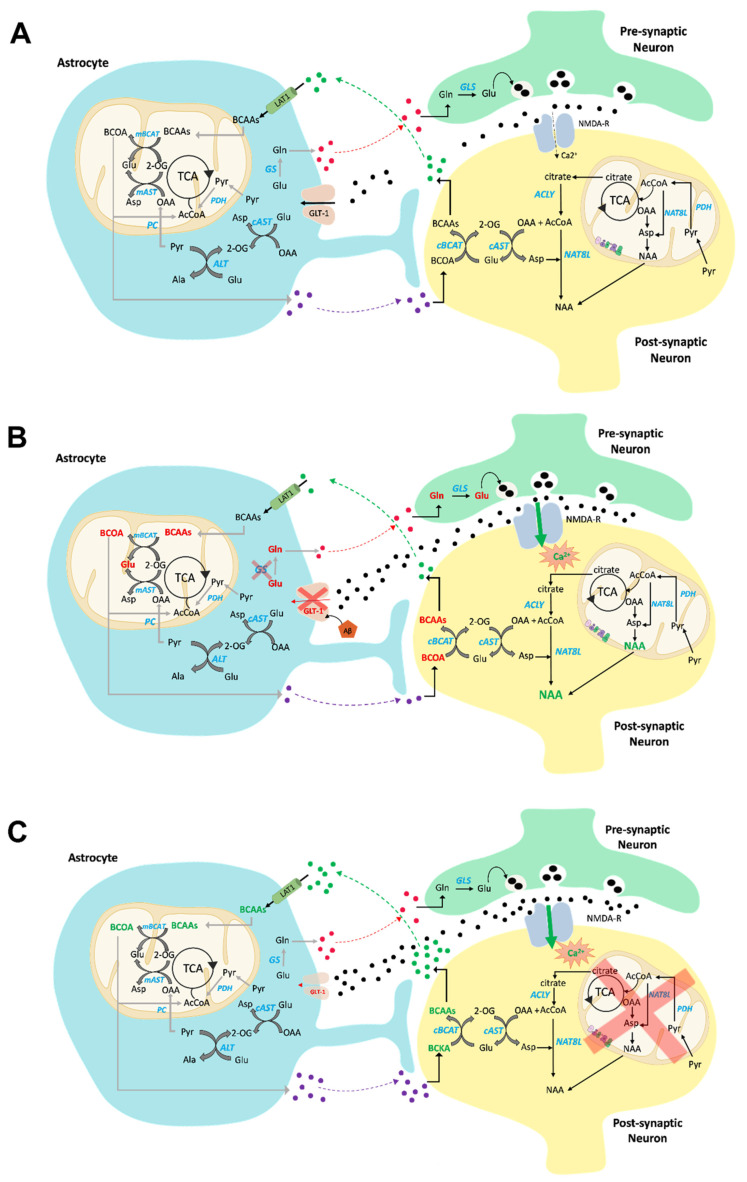
**Overview of amino acid (AA) metabolism in normal (A), AD (B), and PD (C) brains.** (**A**) BCAAs enter the CNS via the BBB (LAT1) and, mainly in astrocytes (mitochondria), undergo transamination, which yields the corresponding BCOAs and glutamate via BCAT. BCOAs enter the TCA for energy production; in this way, BCATs are a constant source of glutamate. Astrocytes then release the oxoacids to the neuron, where they are reconverted to BCAAs (cytosol), which are released back into the extracellular space. Glutamate is an excitatory neurotransmitter. Once glutamate’s signaling role is executed, it is taken up by astrocytes (GLT-1, also known as EAAT2), in which glutamate is converted to glutamine in a reaction catalyzed by GS. This prevents the so-called excitotoxic effect of glutamate accumulation in the synapse. Furthermore, de novo glutamate synthesis occurs exclusively in astrocytes following pyruvate-carboxylase-dependent anaplerosis and BCAA transamination. Glutamate transamination also occurs in the brain as a result of the activity of ALT and AST in both astrocytes and neurons. In this way, ammonia transfer can occur. (**B**) BCAA metabolism is altered in AD brains. Reduced levels of BCAAs (indicated in red) have been found in the blood, CSF, and brains of AD patients, and this reduction is associated with cognitive decline in AD. BCAA diminution might impair glutamate synthesis, leading to impaired neurotransmission and impaired NMDAR function. The glutamate/glutamine cycle is impaired in AD brains, due to GS and GLT-1 oxidation-related loss of activity in astrocytes, exposing neurons to the effect of glutamate excitotoxicity. NAA is accumulated in AD brains, suggesting a cytosolic and mitochondrial metabolic compromise in AD brains. (**C**) BCAA metabolism is altered in PD brains, with accumulation of these amino acids (in green). Glutamate excitotoxicity is also prominent in PD brains, and this is associated with downregulation of astrocytic GLT-1. Increased metabolite levels are shown in green, whereas decreased levels are shown in red.

**Figure 4 molecules-27-00951-f004:**
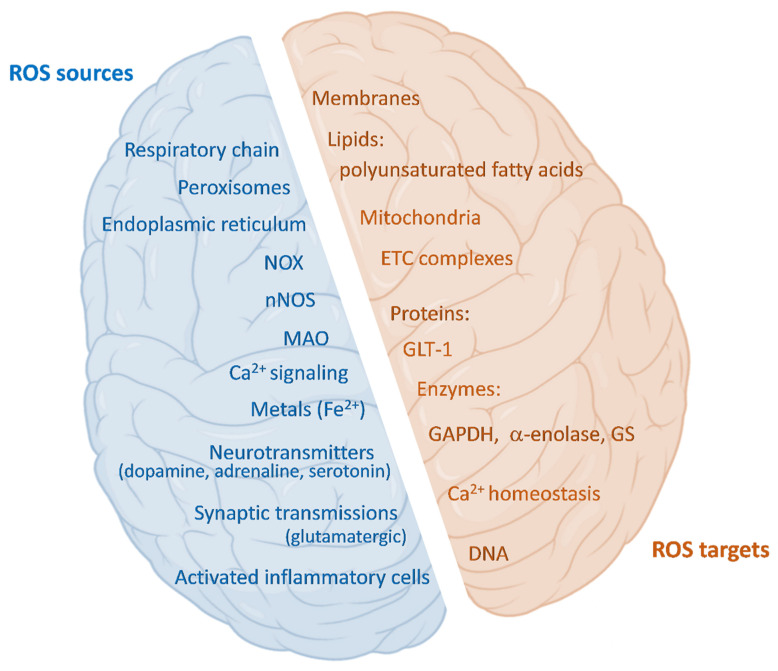
**ROS sources and targets in the brain.** Schematic representation of cerebral ROS sources and targets. Sources: beyond the mitochondrial inner-membrane-resident respiratory chain, peroxisomes, endoplasmic reticulum, NADPH oxidases (NOX), and activated inflammatory cells, there are also specific brain sources such as neuronal nitric oxide synthase (nNOS), monoamine oxidases (MAO), Ca^2+^ signaling, redox-active metal ions, neurotransmitters, and synaptic transmission. Targets of ROS include lipids present in membranes, in particular polyunsaturated fatty acids; mitochondria, and especially the electron transport chain (ETC) complexes; proteins such as the glutamate transporter (GLT-1), or enzymes, for example (among others), glyceraldehyde 3-phosphate dehydrogenase (GAPDH), α-enolase, or glutamine synthetase (GS) in AD brains; Ca^2+^ homeostasis; DNA.

**Table 1 molecules-27-00951-t001:** Correlation between metabolic alterations and clinical features in Alzheimer disease (AD) and Parkinson disease (PD).

Clinical Features	Metabolic Alterations	Pathologies
Vulnerability of hippocampus, lateral and medial temporal lobes, and posterior cingulate/precuneus Reduction in neuronal and synaptic activity Atrophy of the cortical regions Axonal damage Neurodegeneration	Glucose hypometabolism Aerobic glycolysis reduction Elevated levels of lactate and pyruvate Impairment of lactate shuttle Aβ oligomer accumulation Insulin resistance Reduced number of GLUTs Reduced TCA cycle metabolism Reduced activity of the ETC complexes Downregulation of IDH PPP impairment Altered GSH/GSSG ratio Oxidative stress Oxydated GLT1 and GS Glutamate excitotoxicity	AD, PD
Cognitive decline Dementia Impaired neurotransmission	Glucose hypometabolism Aβ oligomer accumulation Insulin resistance Reduced TCA cycle metabolism Reduced activity of the ETC complexes Oxidative stress Reduced Blood BCAAs Affected glutamate synthesis Decreased levels of glutamine Oxydated GLT1 and GS Glutamate excitotoxicity	AD
Insulin desensitization Brain insulin resistance	Glucose hypometabolism Abnormalities in mitochondrial structure and function Aβ oligomer accumulation Secretion of pro-inflammatory cytokines (TNF-α) Oxidative stress Energy deficiency	AD, PD
Chronic inflammation	Downregulation of BDNF and NGF Oxidative stress	AD, PD
Synaptic spine deterioration BBB disfunction	Aβ oligomer accumulation Reduction in the number of plasma membrane insulin receptors Complex IV dysfunction Oxidative stress	AD
Death of dopaminergic neurons Neurodegeneration	Decline of insulin receptors Hyperinsulinemia Insulin resistance GLT1 downregulation Glutamate excitotoxicity Downregulation of metabolism of glycine, serine, and threonine Ornithine and proline accumulation Altered collagen homeostasis	PD
Cell death	Inactivation of PKM2 Downregulation of the Wnt/β-catenin pathway Complex I dysfunction Oxidative stress ATP deficiency	AD, PD
α-Synuclein (α-syn) aggregation	Glucose hypometabolism GAPDH oxydation Abnormalities in mitochondrial structure and function Complex I dysfunction Oxidative stress Reduced ΔΨm	PD
Neurological deficits Hemolytic anemia Myopathy Locomotive defects Loss of DA neurons	Reduced TCA cycle metabolism Accumulation of citrate and 2-OG Epigenetic regulation modifications Deficiency of PGK activity Defective ATP production Defective dopamine production	PD
White matter degeneration Demyelination	Bioenergetic shift from glucose toward ketones Preserved metabolism of ketones Decreased MCT1 expression in the BBB Decline in mitochondrial respiration Oxidative stress Catabolism of myelin lipids into fatty acids to produce ketone bodies	AD, PD

**Table 2 molecules-27-00951-t002:** Therapeutic treatments for Alzheimer’s disease (AD) and Parkinson’s disease (PD), with impacts on metabolism.

Agent	Mechanism of Action	Status	Pathologies
Valproate	Inhibition of tau phosphorylation by targeting glycogen synthase kinase 3 (GSK3β)	Phase II clinical trials	AD [396]
Metformin	Insulin-sensitizing agent	Pilot randomized placebo controlled clinical trial	AD [397]
Intranasal insulin	Increasing the availability of insulin at the brain level	Phase III clinical trials	AD [398]
Monoamine oxidase B (MAOB) inhibitors	Inhibiting the MAO type B, thus enhancing dopamine levels Decreasing oxidative stress	Currently available	PD [399]
Terazosin	Enhancing the activity of phosphoglycerate kinase 1 (PGK1), thereby increasing cellular ATP and dopamine levels	Phase II clinical trials	PD [174]
Nucleotinamide riboside supplementation	Enhancing NAD^+^ biosynthesis	Clinical trials, phase not applicable	PD [400]
Nutritional ketosis	Non-pharmacological treatment Enhancing PPP, GSH levels, and ATP production	Clinical trials, phase not applicable	PD [401]
Mediterranean diet	Non-pharmacological treatment	Observational studies	AD, PD [402]

## Data Availability

Not applicable.

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
