# Peer review of "Metabolic Features of Brain Function with Relevance to Clinical Features of Alzheimer and Parkinson Diseases"

_molecules, 2022, doi:10.3390/molecules27030951_

Round 1

Reviewer 1 Report

The abstract does not reflect the comprehensive review done for the authors. I believe that this section could be improved.

Information in section 2, is very important but this information can be found in many sources. This section might be tedious and distract the readers from other sections more relevant to the present work. Would be great if the authors find a way to make a summary of this information.

The Conclusion´s section must highlight the major findings of the review that might generate new therapeutic options against AD and PD. Also, the authors might propose lines of knowledge generation derived from this work.

Homogenize the fonts in line 6, page 2; line 13, page 11; line 52, page 19; line 2, page 20.

Author Response

Reviewer 1

COMMENT: The abstract does not reflect the comprehensive review done for the authors. I believe that this section could be improved.

RESPONSE: The abstract was modified according to the reviewer’s indications.

COMMENT: Information in section 2, is very important but this information can be found in many sources. This section might be tedious and distract the readers from other sections more relevant to the present work. Would be great if the authors find a way to make a summary of this information.

RESPONSE: The Introduction was slightly modified but not summarized, as we believe that it is useful for the reader to have in the same review an outline of the brain metabolic features in physiological conditions.

COMMENT: The Conclusion´s section must highlight the major findings of the review that might generate new therapeutic options against AD and PD. Also, the authors might propose lines of knowledge generation derived from this work.

RESPONSE: We believe that the Conclusions section has been improved according to the Reviewer’s suggestions.

COMMENT: Homogenize the fonts in line 6, page 2; line 13, page 11; line 52, page 19; line 2, page 20.

RESPONSE: The fonts were homogenized.

Reviewer 2 Report

Please include 1 tables about clinical features of the brain and discuss how metabolic changes are involved in Alzheimer disease and Parkinson disease.

Please include 2 tables about the current clinical approaches in Alzheimer disease and Parkinson disease.

The figure are difficult to follow, please simplify.

Author Response

Reviewer 2

COMMENT: Please include 1 tables about clinical features of the brain and discuss how metabolic changes are involved in Alzheimer disease and Parkinson disease.

RESPONSE: A table is now included in the revised version of the manuscript.

COMMENT: Please include 2 tables about the current clinical approaches in Alzheimer disease and Parkinson disease.

A Table on the current clinical approaches regarding metabolism in AD and PD brain is now included.

COMMENT: The figure are difficult to follow, please simplify.

RESPONSE:  We tried to make the Figure Legends more explanatory of the figures.

Reviewer 3 Report

In this manuscript Butterfield and colleagues reviewed brain metabolism in physilogical states and it's relevance to AD and PD. They discussed metabolic processes of glucose, TCA cycle, ketone bodies, and aminoacids and give  details of their alterations in AD and PD. In general, it is very helpful to understand the metabolic problems in AD and PD.

minor points

mTORC1 pathway is very important in sensing and controlling cellular metabolism and the authors should give a short introduction.

Author Response

Reviewer 3

COMMENT: In this manuscript Butterfield and colleagues reviewed brain metabolism in physilogical states and it's relevance to AD and PD. They discussed metabolic processes of glucose, TCA cycle, ketone bodies, and aminoacids and give  details of their alterations in AD and PD. In general, it is very helpful to understand the metabolic problems in AD and PD.

RESPONSE: We thank the reviewer for her/his remarks about our paper.

COMMENT: mTORC1 pathway is very important in sensing and controlling cellular metabolism and the authors should give a short introduction.

RESPONSE:  A short paragraph on mTOR, including mTORC1, is now included in the revised manuscript (page 6).

Reviewer 4 Report

This is an impressive review about brain, neurodegenerative diseases and metabolism. It will be useful to understand this field and AD/PD pathophysiology. The explanations are well served by the figures

Author Response

Reviewer 4

COMMENT: This is an impressive review about brain, neurodegenerative diseases and metabolism. It will be useful to understand this field and AD/PD pathophysiology. The explanations are well served by the figures

RESPONSE; We thank the reviewer for her/his positive comments of our paper, particularly about the value of the figures.

Round 2

Reviewer 2 Report

Please write the full name in the table legends for AD and PD, in Table-1.

Author Response

We made this minor change as requested by Reviewer 2 in Table 1 and Table 2.